# Climate, Land Use and Land Cover Changes in the Bandama Basin (Côte D'Ivoire, West Africa) and Incidences on Hydropower Production of the Kossou Dam

**Yao Morton Kouame** [1,2], **Salomon Obahoundje** [3,*] , **Arona Diedhiou** [3,4] , **Baptiste François** [5] ,
**Ernest Amoussou** [6], **Sandrine Anquetin** [4] , **Régis Sacre Didi** [3], **Lazare Kouakou Kouassi** [7],
**Vami Hermann N'guessan Bi** [8], **Emile Gneneyougo Soro** [2] and **Etienne Kouakou Yao** [2]

1 Centre de Recherche en Ecologie, Université Nangui Abrogoua, 02 BP 801 Abidjan 02, Côte d'Ivoire
2 Laboratoire Géosciences et Environnement, Université Nangui Abrogoua, 02 BP 801 Abidjan 02,
   Côte d'Ivoire
3 LAPAMF—African Centre of Excellence on Climate Change, Biodiversity and Sustainable
   Development/Université Félix Houphouët Boigny, 22 B.P. 582 Abidjan 22, Côte d'Ivoire
4 IRD, CNRS, Grenoble INP, IGE, Univ. Grenoble Alpes, F-38000 Grenoble, France
5 Department of Civil and Environmental Engineering, University of Massachusetts Amherst, Amherst,
   MA 01003, USA
6 Département de Géographie et Aménagement de Territoire (DGAT/FLASH), Université de Parakou,
   BP 123 Parakou, Bénin
7 UFR-Environnement, Université Jean Lorougnon GUEDE, 12 BP V 25 Daloa 12, Côte d'Ivoire
8 Centre Universitaire de Recherche et d'Application en Télédétection (CURAT), Université Félix
   Houphouët Boigny, 22 BP 801 Abidjan 22, Côte d'Ivoire
* Correspondence: obahoundjes@yahoo.com

**Abstract:** Climate and land use/cover changes are potential drivers of change in hydrology and water use. Incidences of these factors on Bandama hydrological basin and Kossou hydropower generation (1981–2016) in West Africa are assessed in this present work. Using Landsat products of United Stated Geological Survey, results show that water bodies areas and land use have increased by 1.89%/year and 11.56%/year respectively, whereas herbaceous savanna, savanna, forest and evergreen forest coverage have been reduced by 1.39%/year, 0.02%/year, 2.39%/year and 3.33%/year respectively from 1988 to 2016. Hydroclimatic analysis reveals that streamflow presents greater change in magnitude compared to rainfall though both increasing trends are not statistically significant at annual scale. Streamflow varies at least four (two) times greatly than the rainfall (monthly and seasonally) annually except during driest months probably due to land use/cover change. In contrast, Kossou hydropower generation is significantly decreasing (*p*-value 0.007) at both monthly and annual scales possibly due to water abstraction at upstream. Further works are required to elucidate the combined effects of land use/cover and climate changes on hydrological system as well as water abstraction on Kossou generation.

**Keywords:** Rainfall; streamflow; land use; land cover change; hydropower; hydropower generation

---

## 1. Introduction

Increasing share of renewable energy has received considerable attention in recent years due to Paris Agreement. The ambition is to decrease greenhouse gas concentration in the atmosphere attributed to anthropogenic activities such as energy production from fossil fuels and deforestation or conversion

of vegetative areas to agricultural lands [1]. In the tropics and subtropics, deforestation is attributed to large-scale commercial agriculture, subsistence agriculture, infrastructural development, urban expansion and mining [2]. In Africa and according to the Food and Agriculture Organization (FAO), fuelwood charcoal, timber logging, livestock grazing in forest and uncontrolled fires are the main contributors of forest degradation [2]. Within a decade (2000–2010) in Africa, 8000 km$^2$, 7000 km$^2$, 2500 km$^2$, 2000 km$^2$ and 500 km$^2$ of forest were converted every year to commercial agriculture, local agriculture, infrastructure, mining and urban expansion, respectively [2]. Such trends in land cover change highlight the growing pressure on African forest and natural lands. Land use and land cover changes play a role in the change of regional patterns of temperatures, precipitation and vegetation [3,4]. Change in land cover can reduce the amount of carbon sequestrated and can increase the greenhouse emissions; consequently, they can influence the dynamics of local [5] and global [6] climate and hydrological systems [4].

Climate and land cover changes are already affecting the hydrological cycle in West African countries [5,6]. Human-induced greenhouse gas emissions have contributed to the observed intensification of heavy precipitation events [7]. A linear correlation has been found between the normalized difference vegetation index and the precipitation in the Sudanian savanna region [8]. In the Côte d'Ivoire, it has been proved that there is a strong correlation between the changes in vegetation cover and the decrease in rainfall [9], which subsequently impacts hydrological cycles.

Land use and climate are two main factors directly influencing the catchment hydrology [6] and understanding their respective effects is of great importance for land use planning and water resources management [10]. It has been shown that climate change and land development have more impact on changing the seasonal distributions of the streamflow than on altering average annual amounts of the streamflow [11]. For instance, storm runoff extremes increase in most regions at rates higher than suggested by Clausius-Clapeyron scaling, which are systematically close to or exceed those of precipitation extremes over most regions of the globe, accompanied by large spatial and decadal variability due to land anthropogenic change [12]. The changes in land cover/use and/or degradation of the watershed that involved destruction of natural vegetative covers, expansion of croplands, overgrazing and increased area under anthropogenic plantations have resulted partly to an adverse change in observed streamflow [13]. In Sahelian areas, runoff coefficient generally increased along with river discharges induced by decrease in vegetation cover [14]. In Ethiopia, simulation results for the Tekeze dam watershed indicates that increasing bare land and agricultural areas resulted in increased annual and seasonal streamflow and sediment yield in volumes [15]. The hydrological response is more sensitive to land use/cover dynamics [16,17] mainly for the wettest month of the year [18]. Apart from the hydrological responses to land use/cover change, this phenomenon has its implication on water quality through sediment yield [11,16].

The scientific community identified more than a decade ago the importance of land use change on hydrology [19]. Nonetheless, the African continent has lack of sufficient observational data to draw robust conclusions about trends in annual precipitation [20] and its relationship with streamflow and land use/cover changes. Despite this, many studies have been carried out on climate variability across West Africa [21,22]. These studies have shown that a drought trend emerged at the end of the 1960s [23,24] and a return to wet conditions between 1980s and 1990s over West Africa. In Côte d'Ivoire, the results of studies on climatic fluctuations showed a decrease in both surface and underground water resources due to a decrease in rainfall [25,26]. Most studies in West Africa have focused on the impact of climate variability on water resources and agriculture but those focusing on hydropower generation are rare. Hydropower generation relies on available water resources that depends on both climate conditions and land use practices within basin. Currently, West Africa energy sector depends largely on hydroelectric power, which is influenced by climate. Despite the fact that some hydropower plants in the region have failed to deliver the demand due to some extremes climate events (drought and flooding) [27], the dependence of the energy sector on climate is likely to increase due to the current deployment of renewable energy projects, including building of hydropower infrastructures [28].

The aim of this study is two-fold: (i) to assess the trends in rainfall and streamflow in the Bandama basin and their potential impacts on the hydropower generation of Kossou dam; and (ii) to determine the changes in land use and land cover in the basin and their incidences on the hydrological system.

## 2. Study Area

The Bandama basin is located in the Côte d'Ivoire (West Africa). Its catchment area roughly equals 97,000 Km$^2$ and its length 1050 Km. Its climate relies in West Africa climate system which is controlled by the movement of the Inter Tropical Convergence Zone (ITCZ) and influenced by the Monsoon and Harmattan [29]. Harmattan is a very dry, dusty easterly or north-easterly wind on the West African coast, occurring from December to February. When this wind meets the south-westerly humid wind from the Atlantic Ocean (monsoon) it forms the ITCZ which determines the rainy zone.

Bandama basin's hydroclimatic characteristics vary progressively along the latitude (north to southward) [30]. The mean annual rainfall ranges from 1300 to 1000 mm (north to southward) with an average temperature of 26 °C. Variation of temperature within a day can be significant during the hot season with for instance daily maximum being higher than 40 °C while temperature at night can fall below 15 °C. Bandama basin's precipitation is subdivided into three regimes/climates. The first regime is the northern part of the basin located in Sahelian zone with a dry season from November to April and a rainy season from May to October. The second and third regime for the center (sudano-sahelain zone) and southern (Guinean zone) parts of the basin respectively are characterized by four seasons (a long dry season from December to February, a great rainy season from March to June, a short dry season from July to August and the second rainy season from September to November). The second and third regimes are differentiated from their rainfall amount and the start rainy season which is controlled by the migration of ITCZ. Kossou dam is located in between the second and third precipitation regime.

Bandama basin vegetation is composed of three main types namely clear forest and easily penetrable (called forest for next used); inaccessible gallery forest (evergreen forest for the next used) and woody savanna associated with cocoa and coffee plantation (savanna and herbaceous savanna for the next used) [30]. The Sahelian zone (north) and sudano-sahelian are mainly dominated by savanna and forest respectively while the Guinean zone is made of more evergreen forest [31]. However, the Bandama soil is made of hydromorphic soil, eutrophic ferruginous brown tropical soil and granite [30].

The population living in this basin is mainly rural and has rain fed agriculture as their main activity [30]. The agriculture practice in the basin leads to major deforestation responsible for soil erosion. Within the last decade, water consumption in the basin has increased due to activities such as mining and irrigation. In Bandama basin are located two of the four largest hydropower plants of Côte d'Ivoire (namely Kossou and Taabo). This study focuses on Kossou hydropower plant third largest Côte d'Ivoire hydropower plant located between 6°58 N–8°06 N and 5°18 W–5°50 W and the study domain covers an area of 58,700 km$^2$ of Bandama basin with outlet at Taabo hydrological station (Figure 1). Kossou hydropower plant has been initially built for power generation only [32]. Kossou hydropower was established in 1971 with a storage capacity of 27,675 million cubic meter and an installed capacity of 174 Mega Watt (MW) [33].

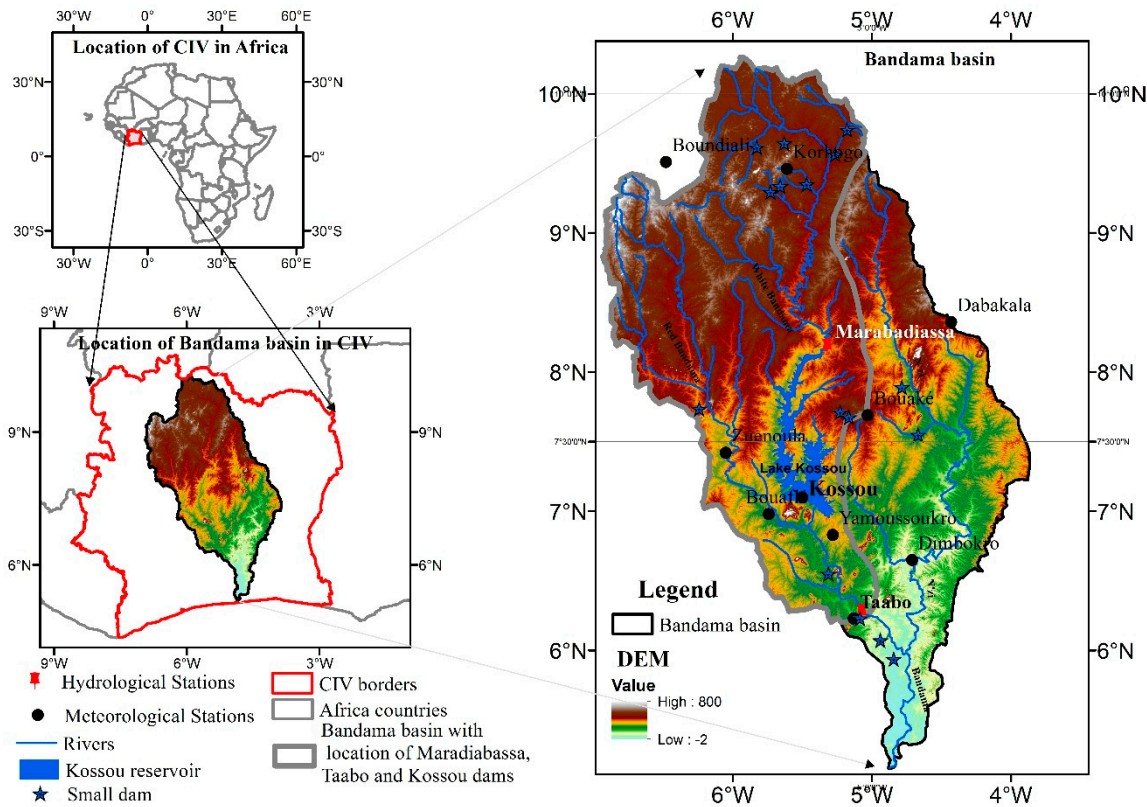

**Figure 1.** Location of Bandama basin.

## 3. Data and Methods

### 3.1. Hydroclimatic and Energy Production Data Sources and Analysis

The hydropower generation of the Kossou hydropower plant (1981–2014) was obtained from the Ivorian Electricity Company. Observed rainfall (rain gauges data for the 1981–2005 period) and streamflow data (1981–2014) were obtained from the Meteorological National Agency and from the National Hydrological Services. However, monthly rainfall data (1981–2014) for the whole Bandama basin was extracted from the Global Precipitation Climatology Project (GPCP) data [34]. The GPCP products are merged data from rain gauge stations, satellites, and sounding observations to estimate monthly rainfall on a 2.5-degree global grid from 1979 to the present [35]. GPCP was used in some previous works in the region [36,37]. The GPCP data was compared with basin mean observed (computed from 10 meteorological stations using Kriging method; see Figure 1) rainfall for the period 1981–2005. We noted that they are strongly correlated (with correlation coefficient = 0.8) with the same trend (Figure 2). The main difference is that GPCP data overestimates the rainfall over the basin in the range of 14.5 mm/year and this difference is not significant at monthly time scale. Similar bias in GPCP dataset has been shown in central part of Africa [38].

The inter-annual variability of GPCP rainfall over Bandama basin and streamflow at Kossou station was investigated (1981–2014) using non-parametric Mann-Kendall test with significance level of 95% [39]. Some basic statistical analysis (standard deviation, mean and coefficient of variation) was performed to assess the link between streamflow and rainfall variability and land cover/use dynamic at annually, monthly and seasonally time scale for the 1981–2014 period.

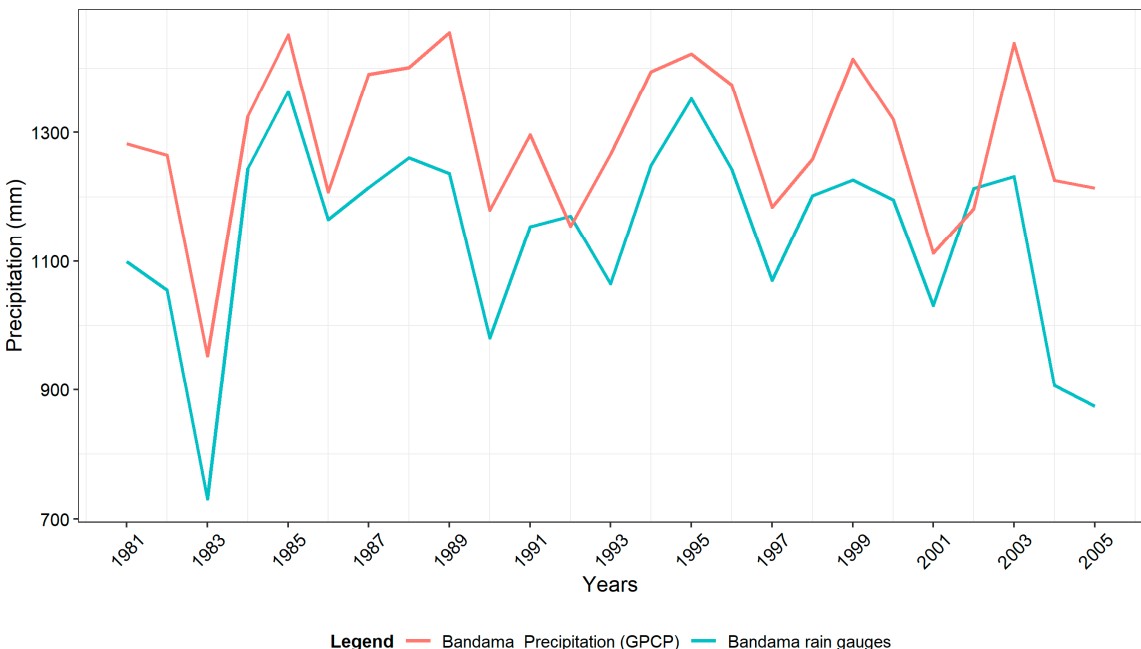

**Figure 2.** Comparison of observed rainfall (rain gauges) and GPCP precipitation over Bandama basin for the 1981–2014 period.

*3.2. Land Use and Land Cover Data Assessment*

To assess the trends in land use and land cover changes, Landsat data from United State Geological Surveys was used: The Landsat Thematic Mapper (TM), the Enhanced Thematic Mapper (ETM) and the Operational Land Imager (OLI) were analyzed for 1988, 2002 and 2016 respectively. All these data were extracted for the driest and cloudless months January and February in order to enhance the Landsat data quality. The Landsat data level 2 has been ordered for atmospheric correction. Bands of the same resolution were composited for land cover classification [40]: the bands 7, 4, and 2 were merged for TM products, and the bands 5, 4, 3, 2 and 1 were combined for ETM products while the bands 7, 5, 4,3,2 and 1 were used for OLI products to get the true value (natural color). The analyses on land use and land cover changes in this study are done with the maximum likelihood pixel-based classification method, which is the most commonly used method for Landsat images [41].

The land cover of the basin has been categorized in six clusters namely water bodies, land use (built-up area and agricultural land), herbaceous savanna, savanna, forest and evergreen forest. Here, "water bodies" are defined as the streamline, small reservoirs, dugouts, lakes and dams. The "land use" cluster refers in one hand to built-up which is made of urbanized areas and the roads or land covered by buildings and other man-made structures, residential, commercial services, industrial area, mixed urban or built-up lands as well as barren land. In another hand, "land use" refers to the agricultural land which represents the farmland areas, or the lands covered with temporary crops followed by harvest period, crop fields and pastures. "Savanna" cluster represents grassland with scattered trees, grading into either open plain or woodland while "herbaceous savanna" refers to grassland. "Forest" cluster refers to trees and other plants covering a large area while the "evergreen forests" are forest made up of rainforest trees in tropical zone or dense forest.

To calibrate and validate the land cover classification the accuracy assessment was performed, and the kappa coefficient was used as the statistical parameter [42]. To statistically quantify errors, a random selection of pixels from ground truth (Google Earth imagery) was compared to classified satellite images as recommended in Ref. [43]. This approach consists of the reclassification of land cover data in another way by following the step described in Foody [43] and Congalton [44] studies and later used by Xu et. [45] and Koglo et al. [46].

The kappa coefficient K, a discrete multivariate technique used in accuracy assessments of thematic maps, is an efficient approach to derive information from an image via the confusion matrix. K > 0.80 represents strong agreement and good accuracy, 0.40–0.80 is middle, and <0.40 is poor [42,47].

## 4. Results

### 4.1. Hydroclimatic Context of Bandama Basin

The inter- and intra-annual (1981–2014) GPCP rainfall (P) data analysis over study basin trends are not statistically significant at 95% confidence level except the month of November. The same conclusion is made at inter-annual for streamflow (Q). However, the intra-annual variability of streamflow is statistically significant at 95% confidence level for the months of February, March, April, October, November and December (Table 1).

**Table 1.** Basic statistics of monthly and annual GPCP rainfall (mm) over Bandama basin and streamflow(m$^3$/s) at Kossou station (1981–2014).

| | Rainfall (P) | | | | Streamflow (Q) | | | |
|---|---|---|---|---|---|---|---|---|
| | Mean(mm) | SD | CV (%) | Score (S) | Mean(m$^3$/s) | SD | CV (%) | Score (S) |
| Jan | 11.67 | 13.11 | 112.36 | 69 | 4.52 | 7.20 | 159.25 | −109 |
| Feb | 42.15 | 20.11 | 47.71 | 89 | 7.69 * | 14.31 | 186.07 | −167 |
| Mar | 95.42 | 28.84 | 30.22 | 35 | 12.64 * | 15.70 | 124.18 | −272 |
| Apr | 127.84 | 22.40 | 17.52 | 35 | 18.70 * | 18.79 | 100.51 | −211 |
| May | 149.02 | 27.19 | 18.24 | −45 | 22.22 | 17.44 | 78.47 | −2 |
| Jun | 172.71 | 31.32 | 18.13 | 99 | 33.11 | 23.03 | 69.57 | −100 |
| Jul | 140.75 | 39.28 | 27.91 | −75 | 55.67 | 39.76 | 71.42 | 14 |
| Aug | 166.82 | 50.95 | 30.54 | −79 | 208.57 | 138.88 | 66.58 | 17 |
| Sep | 178.69 | 39.56 | 22.14 | 33 | 412.92 | 190.43 | 46.11 | 99 |
| Oct | 132.74 | 34.95 | 26.33 | 61 | 282.43 * | 141.06 | 49.94 | 199 |
| Nov | 57.71 * | 22.75 | 39.42 | 137 | 59.95 * | 48.13 | 80.28 | 139 |
| Dec | 21.10 | 14.34 | 67.95 | 43 | 6.22 * | 8.47 | 136.01 | −138 |
| Annual | 1296.62 | 121.34 | 9.36 | 33 | 93.72 | 34.48 | 36.79 | 86 |
| Dry season | 30.16 * | 10.02 | 33.25 | 149 | 23.57 | 16.35 | 69.36 | 89 |
| Wet season | 145.50 | 14.77 | 10.15 | −13 | 130.79 | 50.18 | 38.37 | 83 |

**NB:** * refers to the statistically significant trends at 5% significance level or $p < 0.05$ (95% confidence level). Dry season refers to the months from November to February (months with total monthly rainfall less than 60 mm) and wet season the months (from March to October).

At seasonal time scale, streamflow trend (upward) is not statistically significant for dry (November to February; *p*-value = 0.192, score = 89) and wet (March to October; *p*-value = 0.224, score = 83) seasons which is the case of rainfall at wet season with negative score (*p*-value = 0.85, score = −13). In contrast at dry season, rainfall is statistically significant (*p*-value = 0.028, score = 149). Despite, the negative score of rainfall at wet season, streamflow presents a positive score though both trends are not statistically significant.

Moreover, the change magnitude of streamflow is greater compared to the rainfall using score or Mann-Kendall statistic at annually, monthly time scale as well as wet season. For illustration at monthly resolution, the rainfall scores are lower than the streamflow score for the months of October and November in which the streamflow is statistically significant (upward trend) as well as at annual. Even for some months with downward trend of rainfall the streamflow shows an upward trend (July August).

Some basic statistical analysis namely the standard deviation (SD), the mean and the coefficient of variation (CV) expressed in percentage of rainfall (mm) and streamflow (m$^3$/s) at monthly and annual time scale are also presented in Table 1. The streamflow varies greatly (greater CV) compared to the GPCP rainfall at annual and monthly time scale. The streamflow CV is 36.79% while the rainfall CV is 9.36% annually. At wet (dry) season, streamflow CV is 38.37% (69.36%) while rainfall CV is 10.15%

(33.25%). At the annual time scale, the streamflow varies four times greater than the rainfall and at the monthly and seasonally time scale the streamflow varies at least two times greater than the rainfall except the month of January. The same conclusion was made when using in situ observed rainfall over Bandama basin (1981–2005) with Kossou streamflow. As changes in rainfall in the whole basin are not significant, our hypothesis is that the variations of streamflow described above could be basically due to land use and land cover change. Indeed, it was demonstrated that climate change and land development have more impact on changing the seasonal distributions of the streamflow than on altering average annual amounts of the streamflow [11] mainly in the wettest month of the year [18].

*4.2. Land Use and Land Cover Changes and Their Incidences on Streamflow*

4.2.1. Land Use and Land Cover Changes

The land cover and land use were grouped in six (06) clusters namely the water bodies, land use (built-up and agricultural land), herbaceous savanna, savanna, forest and evergreen forest. The land cover maps of the Bandama basin is presented in Figure 3. The evergreen forest covers the south-western part while the southern part is mainly covered by forest savanna. From the central to the northern part herbaceous savanna and land use are in abundance. The water bodies are located at the reservoirs, small dams and streamflow locations. The land use is increasing in detriment of vegetative area (savanna, herbaceous savanna, forest and evergreen forest), which hereafter will be referred to as vegetative area for short.

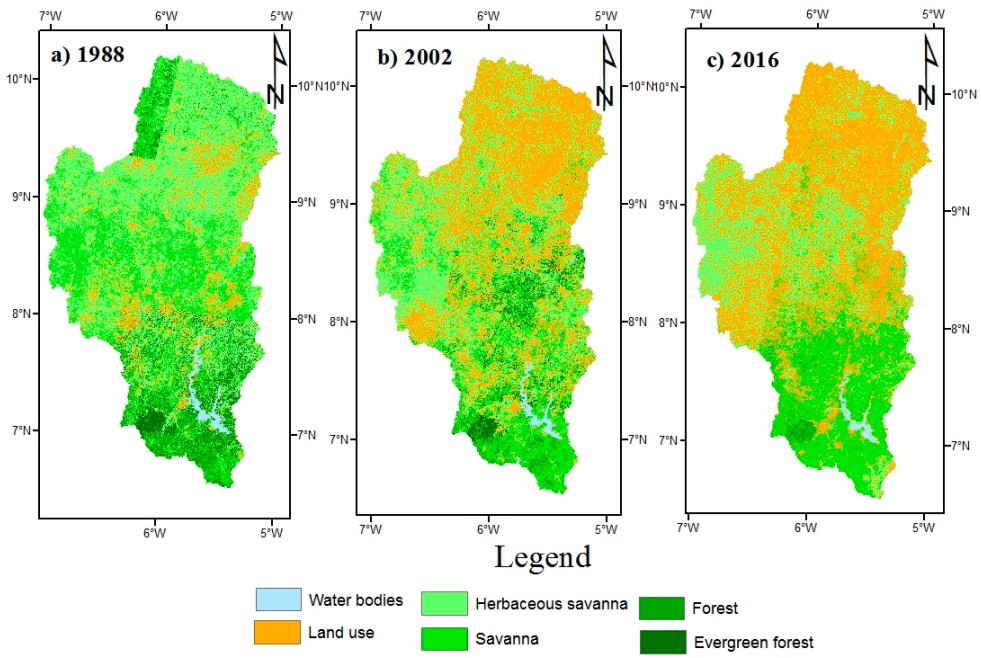

**Figure 3.** Bandama Basin land use and land cover maps.

The trend analysis for the periods 1988–2002, 2002–2016 and 1988–2016 was assessed. The results show an increase in water bodies, land use (built-up areas and agricultural land) while the vegetative area (herbaceous savanna, savanna and the evergreen forest) are decreasing during all the period considered (Table 2). The savanna cluster presents an increasing trend during 2002–2016 period probably due to agroforestry practice in the basin. The basin experienced an increasing trend in forest for the first period and a decreasing for the second one. Generally, for the period 1988–2016, the basin presented a rapid increasing trend of water bodies (1.89%/year), land use (11.56%/year) while a decrease is noted for the herbaceous savanna (−1.39%/year), the savanna (−0.02%/year), the forest (−2.39%/year) and evergreen the forest (−3.33%/year).

**Table 2.** Land use land cover change trend analysis.

| Land Cover Types | Area (%) 1988 | Area (%) 2002 | Area (%) 2016 | 1988–2002 | | 2002–2016 | | 1988–2016 | |
|---|---|---|---|---|---|---|---|---|---|
| | | | | Change (%) | Change Rate (% /year) | Change (%) | Change Rate (% /year) | Change (%) | Change Rate (% /year) |
| Water Bodies | 1.26 | 1.63 | 1.97 | 29.36508 | 1.98 | 20.8589 | 1.39 | 56.34921 | 1.89 |
| Land use | 9.43 | 35.39 | 42.15 | 275.2916 | 18.34 | 19.10144 | 1.27 | 346.9777 | 11.56 |
| Herbaceous Savanna | 57.72 | 42.43 | 33.63 | −26.49 | −1.77 | −20.74 | −1.38 | −41.736 | −1.39 |
| Savanna | 21.22 | 15.22 | 21.12 | −28.2752 | −1.89 | 38.76478 | 2.58 | −0.47125 | −0.02 |
| Forest | 3.62 | 1.29 | 1.14 | −64.3646 | −4.29 | −11.6279 | −0.79 | −68.5083 | −2.29 |
| Evergreen Forest | 6.74 | 4.04 | 0 | −40.0593 | −2.67 | −100 | −6.67 | −100 | −3.33 |

The classified data presents a strong accuracy and Kappa coefficient for the years 1988, 2002, and 2016. The Kappa coefficient and accuracy assessment (Table 3) are computed from confusion matrix described in Foody work [43] and widely used. The confusion matrix of classified data of 1988, 2002 and 2016 years are presented in appendices (Tables A1–A3).

**Table 3.** Kappa Coefficient and Accuracy Assessment of the land cover classification.

|  |  | 1988 | 2002 | 2016 | Qualification |
|---|---|---|---|---|---|
| Bandama | Kappa Coefficient | 88.33% | 92.% | 90% | strong |
|  | Accuracy Assessment | 90.28% | 93.33% | 90% | Strong |

4.2.2. Incidence of Land Use/Cover Changes on Streamflow

Table 4 presents the annual rate of changes in land cover types and the coefficient of variation (CV) express in percentage of rainfall (mm) or streamflow ($m^3$/s) at annual time scale. The streamflow varies greatly than rainfall for all considered periods. The highest variation was observed with greatest decreasing in vegetative coverage mainly evergreen forest during 2002–2014 period. During that period the streamflow varies at least three times than the rainfall. Thus, the conversion of vegetative coverage to land use could be a contributor to this observed variation and increasing trend of streamflow despite no significant trend of rainfall is noted.

**Table 4.** Incidence of land use/cover change on streamflow at Kossou station (1981–2014).

|  |  | Periods | | | Observation |
|---|---|---|---|---|---|
|  |  | **1988–2002** | **2002–2014** | **1988–2014** | |
| **CV (%)** | Rainfall (P) | 8.8 | 8.64 | 8.55 | Less variation |
|  | Streamflow (Q) | 30.05 | 31.14 | 30.54 | High variation |
| **Change %/year (land cover/use classes)** | Land use | 18.34 | 1.27 | 11.56 | Increase in land use |
|  | Herbaceous Savanna | −1.77 | −1.38 | −1.39 | |
|  | Savanna | −1.89 | 2.58 | −0.02 | Decline in vegetative coverage |
|  | Forest | −4.29 | −0.79 | −2.29 | |
|  | Evergreen Forest | −2.67 | −6.67 | −3.33 | |

*4.3. Incidences of Land Use/Cover and Climate Changes on the Evolution of Hydropower Generation in the Kossou Dam*

The annual trends of rainfall, streamflow and hydropower generation are presented in Figure 4. Rainfall and streamflow are increasing though they are not statistically significant while the hydropower generation is declined (Figure 4). The precipitation presents lower magnitude change compared to streamflow. The downward trend of hydropower generation is statistically significant at 95% confidence level (tau =−0.325; *p*-value =0.007).

Figure 5 illustrates the monthly hydropower generation from Kossou dam, the monthly GPCP rainfall over the whole Bandama basin and of the monthly inflow to Kossou dam for three typical dry years (1983, 1992, 2007 in Figure 5A) and three wet years (1985, 1995, 2014 in Figure 5B) are presented in Figure 5. Indeed, rainfall anomaly from GPCP data was computed using Lamb definition [48].

Generally, the analysis shows a decrease in hydropower generation from October to July while it increases from August to September. Hydropower generation follows the river streamflow pattern. The first rainy season as defined earlier contributes to saturate the soil water content capacity and to recharge the reservoir. At the end of the first rainy season (March-July) the reservoir is full. Then the water is released mainly from July to October leading to a greater hydropower generation. In the remaining months before the onset rainfall, the release of the water will be reduced in order to be able to cover water demand for dam operation. This consequently results in a decline of power generation from October to July.

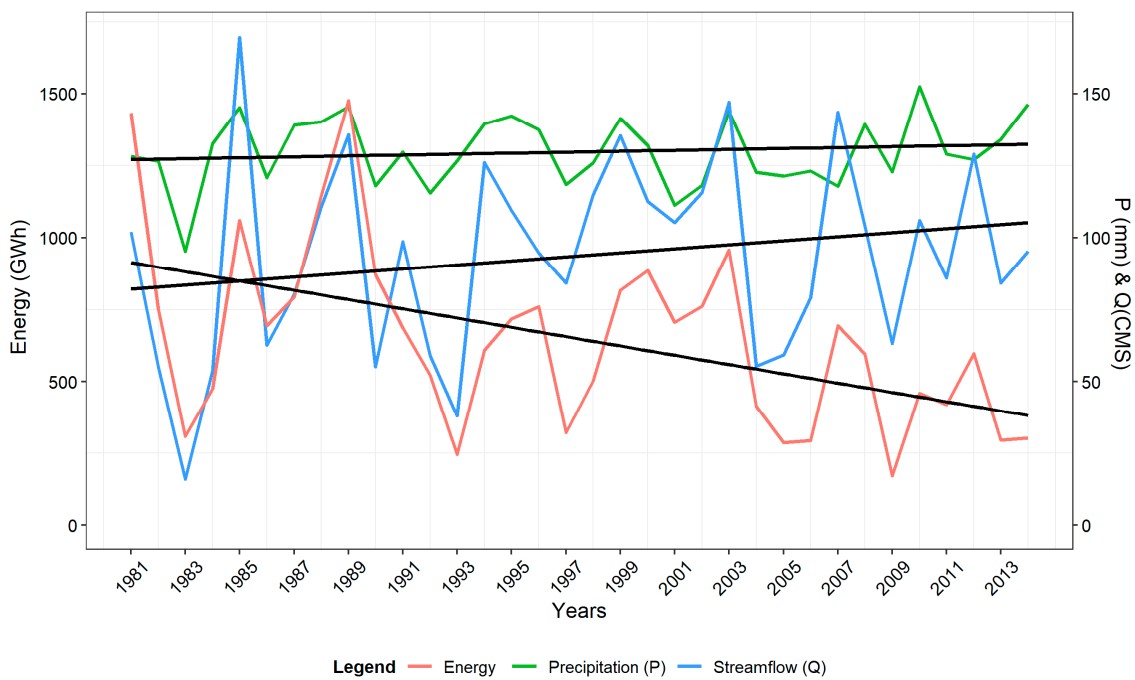

**Figure 4.** Annual variability: total hydropower generation (GWh in red color); mean annual GPCP precipitation (P in mm in green color) and mean annual streamflow (Q in m$^3$/s in blue color). The black line is the regression line for each variable.

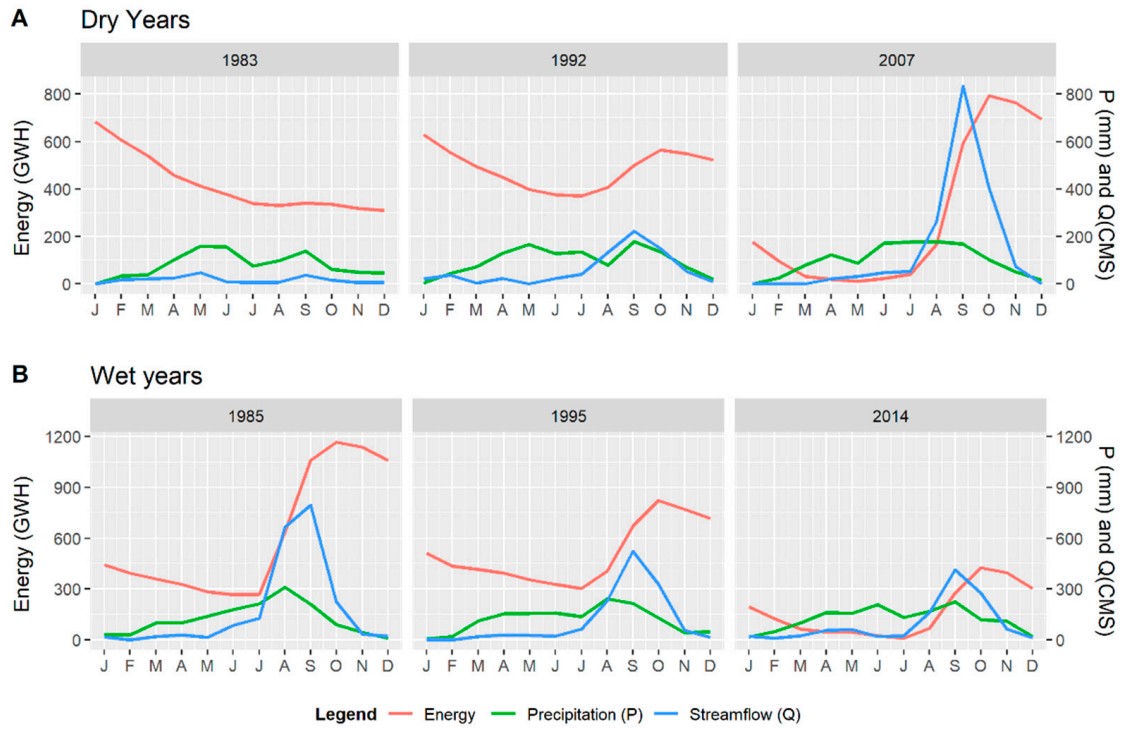

**Figure 5.** Monthly variation of total hydropower production (GWh in red), total GPCP precipitation (P in mm in green) in the basin and mean streamflow (Q in m$^3$/s in blue) at Kossou hydrological station for three dry years (1983, 1992, 2007) and three wet years (1985, 1995, 2014).

The hydropower generation in the chosen wet years are greater compare to the generation in dry years. However, we notice that the difference in total rainfall between wet and dry years is weaker than

the difference in the streamflow magnitude. However, it is worth noting that streamflow increases sharply during the last dry years (1992, 2007 in Figure 5A) despite the lowest magnitude of monthly mean rainfall. This contrast in trend and magnitude of hydropower generation and rainfall could be attributed to an increase in water abstraction at upstream of the hydropower plant. This is justified by the presence of important number of small reservoirs (Figure 1) mainly for irrigation, mining, livestock breeding activities.

Additionally, the annual hydropower generation varies according to the rainfall of previous years and to the distribution of rainfall within the year. For instance, the largest power generation observed at the first months of dry year 1983 (Figure 5A) is justified by the fact that 1983 year follows some normal years. Lower power generation at the first months (January to June) in 1985 and 1995 (Figure 5B) for example could be explained by the fact that they follow 1983 and 1992 dry years respectively affecting the power generation of the next following year. Moreover, the hydropower production presents a sharp decreasing trend during 1983 dry year can also be explained by the weak rainfall amount in the year as well as its distribution. As a small amount of water was stored in the extremely dry year 1983, the production of energy was not possible in 1984 (not shown) until the start of rain season of year 1985. The consecutive dry years 1991, 1992 and 1993 have negatively affected the energy production of the wet years 1994 and 1995 (Figure 5B).

From 2007 to 2014 (Figures 4 and 5A,B), the trend of hydropower generation is decreasing at both inter and intra-annual. Despite the high amount of rainfall in 2014 (Figure 5B) and the previous years which were quite normal, the power generation of 2014 is lower compared to the production of other wet years and dry years. This is due to the reduction in Kossou reservoir water level/filling rate (not shown) resulting in reservoir area reduction [49]. The land use dynamic maps have shown that water bodies are increasing in the basin due to small dams and reservoirs for irrigation which may justify the decline in water level and the downward trend of hydropower generation. The rainfall anomaly over Bandama basin is presented in Appendix B.

Furthermore, as the basin is under pressure of different land use practice, namely the conversion of vegetative area to agricultural land and built-up; the increase in land use and land cover changes could lead to silting and the sedimentation of reservoir. This land use change effects on reservoir could also be assess deeply. Under this condition reservoir capacity could reduce, consequently reducing amount of water stored and energy produced. However, this hypothesis of reservoir reduction associated with sediment transport and silting as result of land use and land cover change and leading to de decrease of hydropower generation needs to be investigated further.

Hydropower generation (Figure 6a,b) is strongly correlated with reservoir water level (WL) and streamflow (Q). The hydropower depends on streamflow and water level of the reservoir. However, Water level is not correlated with both streamflow and precipitation (Figure 6d&e). Despite the strongest correlation of streamflow and precipitation (Figure 6f), the energy generation is not correlated with precipitation (Figure 6c). This may be due to LULCC and water abstraction in upstream of the dam.

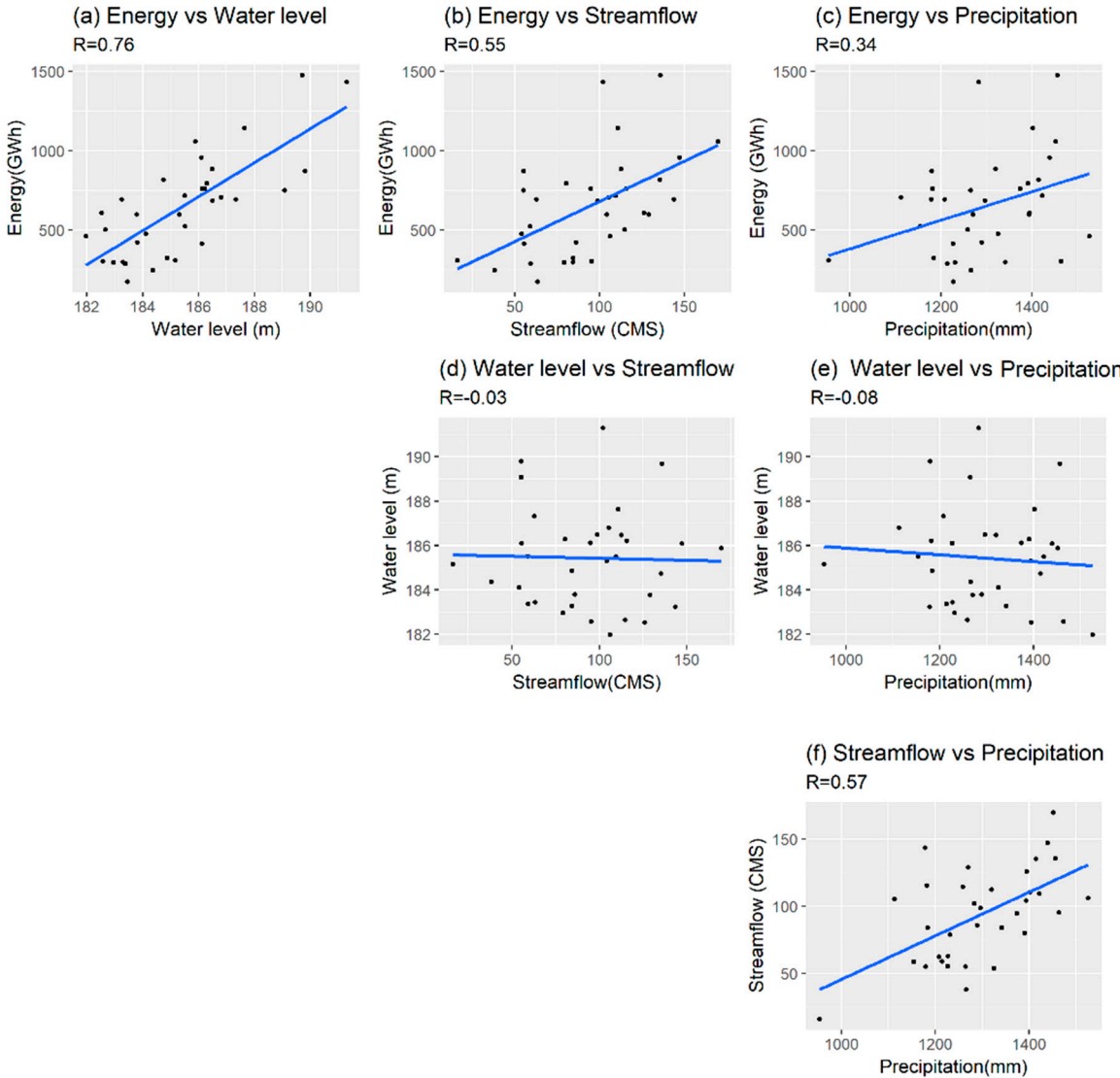

**Figure 6.** Scatter plot between study variables (1981–2014): top row (E = f(WL, Q,P)); middle raw (WL = f(Q,P)) and bottom raw(Q = f(P)).

## 5. Discussion

This study shows that trends in annual rainfall in the Bandama basin and in annual streamflow at Kossou station from 1981 to 2014 are not statistically significant at 95% level of confidence. At monthly time scale, the rainfall trend is statistically significant only in November while the streamflow is for February, March, April, October, November and December months. At seasonally scale, though the rainfall downward trend is not statistically significant during wet season (from March to October), the stream flow presents an upward trend which is not significant too. The magnitude change of streamflow is greater compared to the rainfall one. in addition, streamflow varied much more than rainfall at all scale. This difference in trend, magnitude and variation could be due to land cover/use dynamic in the basin as found by Bewket [13].

The analysis of land cover changes in the basin between 1988 and 2016 reveals that water bodies and land use (built-up and agricultural land) have increased while the vegetative area (savanna, herbaceous savanna, forest and evergreen forest) has decreased. These results are in accordance with previous studies in the region [42,46,50]. The increase in water bodies is due the increase in the construction of small reservoirs, dugouts and dams within the basin. West African basins have received more attention in term of small reservoirs and dugouts, domestic water supplies and for livestock

breeding as well as reservoirs for hydropower plants. The increase in water bodies in the basin is one of main cause of declining in reservoir filling rate. West Africa has rapid population growth rate associated with an increase in urbanization. This has led to the rapid growth in built-up areas. In addition, the increase in population growth rate has also led to an increase in food consumption and food production, explaining the expansion of land use, and coherent with a decline of the vegetative areas (savanna, herbaceous savanna, forest and evergreen forest) [42,46]. Furthermore, the increase in land use could add pressure on water availability by increasing water withdrawal for irrigation, mining and livestock breeding.

Land use and land cover changes are known to influence the climate system [3]. Afforestation option is predicted to increase the rainfall over West Africa [51] while deforestation will reduce it [52]. Some study have shown how the change in vegetation affect the rainfall in the region [8,51]. Land use/cover change has an implication on hydrometeorological system [6] as well as in water balance [53]. Land cover plays an important role in climate system of a basin as well as in runoff generation and rainwater infiltration rate. It has been demonstrated that the less vegetative a land is, higher the runoff generation and weaker the infiltration rate [42]. The results indicate that this might be the case in the Bandama basin. The study shows that at the monthly and seasonally (annual) time scale, the streamflow varies at least two times (four times) greatly than the rainfall except for the driest months December and January. This could be due to land use/land cover change in the basin. Soro et al. (2013) have also concluded that the modifications of land cover has led to changes on hydrological behavior of upstream of Bandama catchment [30]. The annual runoff change is highly sensitive to land use change. Some literature in the region [42,47] and elsewhere [13] confirmed that the relative hydrological effects of forest changes and climatic variability are largely dependent on the magnitude of the change and watershed characteristics. Sahelian rivers in West Africa present an increase in runoff coefficient despite the increase in the number of dams [54] and this is attributed to land use practice [55]. It has been demonstrated that such Sahelian basin has loss their soil water hold capacity [55] and causes large increase in global storm runoff extremes driven by climate and anthropogenic changes [12]. The same phenomenon is the cause of the Sahelian paradox [14,22], which results in increase in streamflow despite the decrease in rainfall over Niger basin.

However, land use land and cover change threats agricultural lands by erosion or physical deterioration, land degradation or salt deposits and loss of micro/macro fauna and flora, as well as declines in soil fertility up to 32.5% and soil water holding capacity up to 11.7%, and changes in soil texture up to 3.3% [46]. This contributes to the strong reduction of crops yield and make farmers to shift towards new lands more fertile or to adopt climate smart practices, with an emphasis on cost-effective drip irrigation systems and other modern practice [46]. This could add pressure on water availability for hydropower generation and create conflict among water users' sectors within country and among countries. This may be the case of Bandama basin too. Furthermore, the decline in vegetative cover associated with an increase in land use could lead to low underground recharge and high runoff generation during the rainy period. The Bandama basin underground recharge water is decreasing due to the land use and land cover change associated with changing climatic conditions [56]. Consequently, this could contribute to the increase of river discharge [30] as showed in Chemoga river in Blue Nile basin [13] and in Black Volta basins [42,47]. The hydrological dynamic of basin is more sensitive to land use/cover dynamic [57] and this also influence the sediment yield of the river [11,16]. Nevertheless, the combined effect of climate change and land development have more impact on changing the seasonal distributions of the streamflow [58] and nitrogen load than on altering average annual amounts of the streamflow and nitrogen load [11].

Consequently, this increase in river discharge could also affect positively the hydropower generation. For instance, Obahoundje et al. (2017) showed that the increasing conversion of vegetative area to agricultural land and built-up has led to an increase of streamflow and has favored Bui hydropower generation in Ghana [42]. In Tekeze dam catchment (Ethiopia) the mean annual streamflow increased by 6.02% due to land use/cover changes [15]. However, it has been demonstrated that land use/cover

dynamics plays an important role in sediment yield [59] and in water quality of a basin [11,16,17,60,61]. Thus, the change in land cover combined with change in rainfall could lead to increase soil erosion [62] and could result into siltation and sedimentation of rivers as well as of the dam's reservoir volume. Added to the water withdrawal at upstream, this may explain the declined observed in reservoir area which initially was estimated at 1500 km$^2$ for a minimum coastline of 203 m and presently, the current average area of lake Kossou is about 900 km2 with an average coastline of 184 m according to the Food and Agriculture Organization of the United Nations [49]. However, this need to be further investigated in order to quantify the resultants effects of land use/cover change on Kossou hydropower plant, and use of GIS-based research model such as WetSpass (Water and Energy Transfer between Soil, Plants and Atmosphere under quasi-Steady State) model will help to better compute the components of the water balance [63] and to better capture hydrological impacts of climate/land use changes [64,65].

## 6. Conclusions

This study assessed the climate, land use and land cover changes in the Bandama basin and their impacts on hydrological system and on the hydropower generation of Kossou dam for the period 1981–2014. The results show that trends (upward) at inter-annual time scale of rainfall and streamflow are not statistically significant. At intra-annual timescale, the streamflow presents a statistically significant positive trend for the months of February, March, April, October, November and December, while rainfall is statistically significant on the month of November. Despite rainfall exhibits a downward trend during wet season while the streamflow shows an upward trend though they are not statistically significant. In addition, a statistical analysis reveals that at annual (monthly and seasonally) time scale, the streamflow varies at least four (two) time greatly than the rainfall and at least three times at monthly time scale except the driest month January. Moreover, the change magnitude of streamflow is greater than rainfall. This difference in magnitude, trend and variation could be attributed to land use/cover change.

Despite, the observed upward trends of streamflow and rainfall though they are statistically significant, the hydropower generation at Kossou dam exhibits a statistically significant downward trend. This contrast in events is attributed in one hand to water abstraction at upstream of the dam for different land use practices (irrigation or mining activities).

Indeed, the studied basin as well as the area around reservoir are under pressure of different land use practices. The results show that land use (built-up, barren, agricultural land) and water bodies are increasing while the vegetative areas (savanna, herbaceous savanna, forest and evergreen forest) are declining. This change in land cover could alter the hydrological of the basin.

Land use change (conversion of vegetative area to agricultural land and urbanization) could result into increase in the streamflow, which is favorable for hydropower generation which is without consequences on plant system. It is urged to carry out a deep study in this basin in order to rank the degree of contribution of each factor (water abstraction, climate and land cover/use changes), their potential consequences on hydrological and hydropower system of the basin. Thus, a new GIS-based research is planned in the future to quantify water balance components of the basin. Lastly, the managers of dams and river basin authorities should carefully carry out studies on the potential impacts of land use and land cover changes on existing and projected hydro power plants in the future under changing climatic conditions.

**Author Contributions:** Y.M.K. and S.O. conducted this research. A.D. and E.A. were the advisors: Conceptualization, Y.M.K., S.O., R.S.D., A.D. and E.A.; Methodology, Y.M.K., S.O., A.D., E.A., R.S.D. and S.A.; Data Curation, Y.M.K., S.O., L.K.K., V.H.N.B., E.G.S. and E.K.Y.; Writing-Original Draft Preparation, Y.M., S.O., A.D., E.A. and S.A.; Writing-Review & Editing, S.A., B.F.; Supervision, A.D. and E.A.; Funding Acquisition, A.D.

**Funding:** The research leading to this publication is co-funded by the NERC/DFID "Future Climate for Africa" program under the AMMA-2050 project, grant number NE/M019969/1 and by IRD (Institut de Recherche pour le Développement; France) grant number UMR IGE Imputation 252RA5". The support for the final improvement with new additional data and analyses was possible thanks to the Dan David Prize.

**Acknowledgments:** The authors thank the Institute of Research for Development (IRD, France) and Institute of Geosciences for Environment (IGE, University Grenoble Alpes) for providing the facility (the Regional Climate

Modelling Platform) to perform this study at the University Felix Houphouet Boigny (Abidjan, Côte d'Ivoire) and the IT support funded by IRD/PRPT contract.

**Conflicts of Interest:** The authors declare no conflict of interest.

## Appendix A

The classified Landsat images can be subjected to some errors. A statistical approach to quantify these errors is the random selection of pixels from the classification map to be compared to the reference map which produces a confusion matrix. Ten pixels times the number of land cover classes were randomly selected (eg: In 1988, $10 \times 6 = 60$ pixels were selected for each land cover class). The confusion matrix has been widely used for accuracy assessment of the land use land cover maps [30,47]. The main statistical information derived from the confusion matrix are: overall accuracy, commission error, omission error, the producers accuracy, the users accuracy and Kappa Coefficient K described in [43] study. The confusion matrices of the classified images are presented in Tables A1–A3 for 1988, 2002 and 2016 respectively.

**Table A1.** Confusion matrix of the land use/cover classification for 1988.

| 1988 | | Ground Truth (Google Earth imagery) | | | | | | Number of Classified Pixels | User's Accuracy | Commission Error |
| --- | --- | --- | --- | --- | --- | --- | --- | --- | --- | --- |
| | | Water | Land Use | Herbaceous Savanna | Savanna | Forest | Evergreen Forest | | | |
| Classified Satellite image as: | Water | 60 | 1 | 0 | 0 | 0 | 0 | 61 | 98.36 | 1.64 |
| | Land use | 0 | 54 | 0 | 0 | 0 | 0 | 54 | 100.00 | 0.00 |
| | Herbaceous Savanna | 0 | 0 | 50 | 3 | 0 | 0 | 53 | 94.34 | 5.66 |
| | Savanna | 0 | 0 | 10 | 52 | 6 | 0 | 68 | 76.47 | 23.53 |
| | Forest | 0 | 0 | 0 | 5 | 49 | 0 | 54 | 90.74 | 9.26 |
| | Evergreen Forest | 0 | 5 | 0 | 0 | 5 | 60 | 70 | 85.71 | 14.29 |
| Number of ground truth | | 60 | 60 | 60 | 60 | 60 | 60 | 360 | | |
| Producer's Accuracy | | 100.00 | 90.00 | 83.33 | 86.67 | 81.67 | 100.00 | | **Total Accuracy** | **90.28%** |
| Omission error | | 0.00 | 10.00 | 16.67 | 13.33 | 18.33 | 0.00 | | **Kappa Coefficient** | **88.33%** |

**Table A2.** Confusion matrix of the land use/cover classification for 2002 year.

| 2002 | | Ground Truth (Google Earth Imagery) | | | | | | Number of Classified Pixels | User's Accuracy | Commission Error |
| --- | --- | --- | --- | --- | --- | --- | --- | --- | --- | --- |
| | | Water | Land Use | Herbaceous Savanna | Savanna | Forest | Evergreen Forest | | | |
| Classified Satellite image as: | Water | 54 | 0 | 0 | 0 | 0 | 0 | 54 | 100.00 | 0.00 |
| | Land use | 6 | 60 | 0 | 0 | 0 | 0 | 66 | 90.91 | 9.09 |
| | Herbaceous Savanna | 0 | 0 | 42 | 0 | 0 | 0 | 42 | 100.00 | 0.00 |
| | Savanna | 0 | 0 | 18 | 60 | 0 | 0 | 78 | 76.92 | 23.08 |
| | Forest | 0 | 0 | 0 | 0 | 60 | 0 | 60 | 100.00 | 0.00 |
| | Evergreen Forest | 0 | 0 | 0 | 0 | 0 | 60 | 60 | 100.00 | 0.00 |
| Number of ground truth | | 60 | 60 | 60 | 60 | 60 | 60 | 360 | | |
| Producer's Accuracy | | 90 | 100 | 70 | 100 | 100 | 100 | | **Total Accuracy** | **93.33%** |
| Omission error | | 10 | 0 | 30 | 0 | 0 | 0 | | **Kappa Coefficient** | **92.00%** |

Table A3. Confusion matrix of the land use/cover classification for 2016 year.

| 2016 | | Ground Truth (Google Earth Imagery) | | | | | Number of Classified Pixels | User's Accuracy | Commission Error |
|---|---|---|---|---|---|---|---|---|---|
| | | Water | Land Use | Herbaceous Savanna | Savanna | Forest | | | |
| | **Water** | 50 | 0 | 0 | 0 | 0 | 50 | 100 | 0 |
| Classified | **Land use** | 0 | 45 | 0 | 0 | 0 | 45 | 100 | 0 |
| Satellite image | **Herbaceous Savanna** | 0 | 0 | 40 | 10 | 0 | 50 | 80 | 20 |
| as: | **Savanna** | 0 | 0 | 10 | 40 | 0 | 50 | 80 | 20 |
| | **Forest** | 0 | 5 | 0 | 0 | 50 | 55 | 90.91 | 9.09 |
| Number of ground truth | | 50 | 50 | 50 | 50 | 50 | 250 | | |
| Producer's Accuracy | | 100 | 90 | 80 | 80 | 100 | | **Total Accuracy** | **90%** |
| Omission error | | 0 | 10 | 20 | 20 | 0 | | **Kappa Coefficient** | **87.5%** |

## Appendix B

The rainfall (GPCP precipitation) anomaly computed from 1981 to 2014 using Lamb definition [48] is presented in Figure A1.

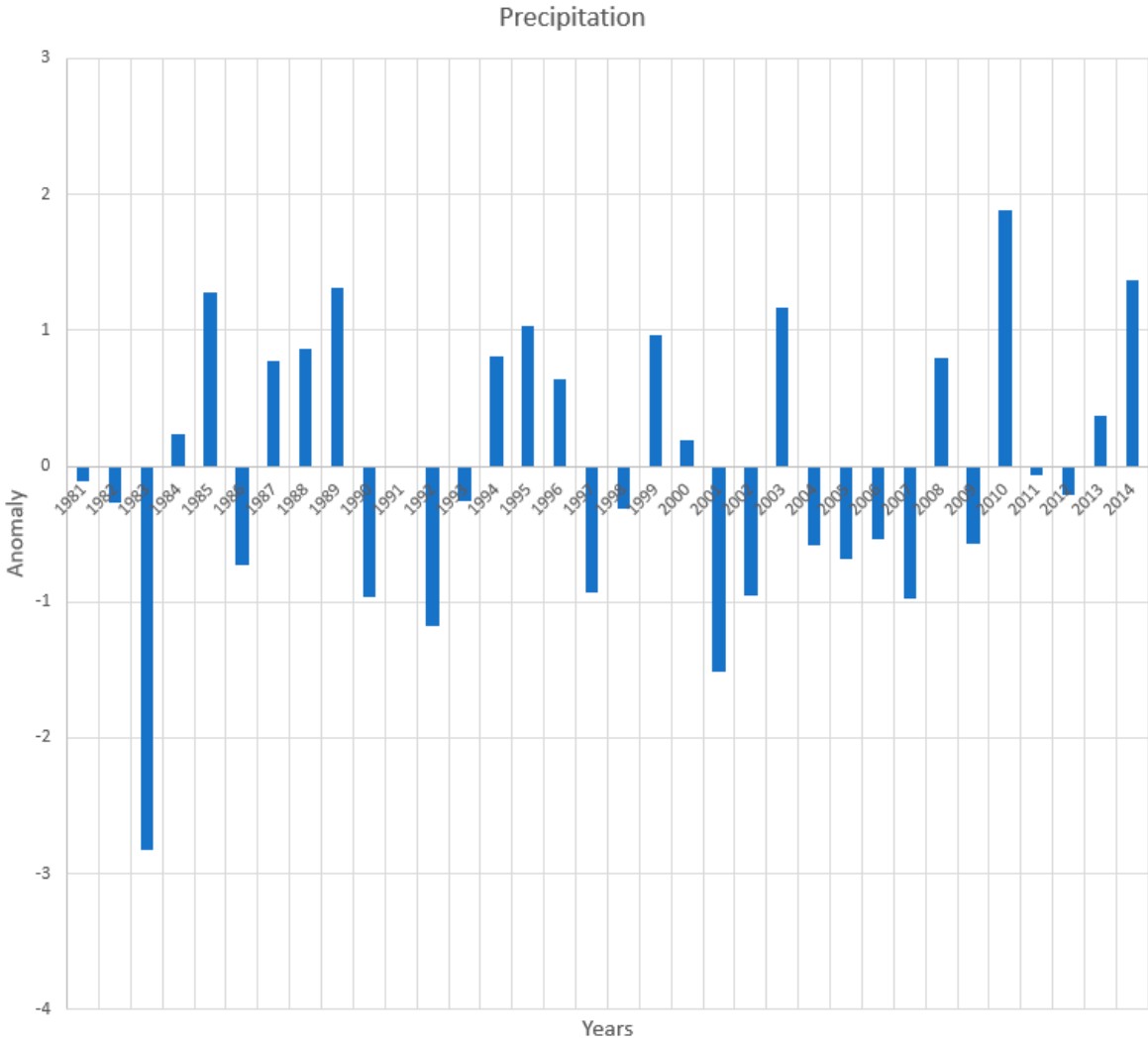

**Figure A1.** Rainfall (GPCP) anomaly over Bandama basin.

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
