# Peer review of "Climate, Land Use and Land Cover Changes in the Bandama Basin (Côte D’Ivoire, West Africa) and Incidences on Hydropower Production of the Kossou Dam"

_land, doi:10.3390/land8070103_

Round 1
Reviewer 1 Report
The authors address a timely and interesting topic as climate, land use and land xover changes impacts on the hydroelectric power generation of the Kossou dam. But before its final acceptation by Land, I suggest some clarification and revision to the authors.
Line 134-136: Why do you use the GPCP rainfall data ? The GPCP has the coarse resolution(2.5 °). How many GPCP grids in your basins? You can also present the grid in Figure 1.
Line 136-139: How many stations did you use to check the precision of GPCP? You should present the results in a Figure.
Line 200: Why did you list the P-value in Table 2? I think you should give the trend value and the confidence level can be showed as “*” …
Line 232-233: Please list some reference to support your opinion.
Figure 3: Unified the value range of Y-axes in wet years
Some small errors: Please check the whole manuscript to correct them. Some examples are listed:
Line 121: “58700 km2”
Line 155: bands 5, 4,3,2
Line 206: m3/s
Line 174: was compare
……
Author Response
Dear Reviewer,
Thank you very much for your time. Please, find attached the response to your question.
Best regards.

Reviewer 2 Report
Evaluation of the manuscript entitled:
Climate, land use and land cover changes in the Bandama Basin (Côte d’Ivoire, West Africa) and incidences on hydropower production of the Kossou’s dam
This article deals with climate, land use and land cover changes and incidences that likely to have on hydropower production in an area in Côte d’Ivoire, West Africa. The issue sounds interesting, but the document has many weaknesses. The authors must answer the next to be able to accept for publication.
1. The abstract needs a qualitative and quantitative strengthen so that it can stand alone for someone who needs to take an idea from your research. Also, the abstract should be a total of about 200 words maximum in accordance with the journal's guidelines.
2. The introduction to the paper is useful as a reference but there is no clear explanation on why the papers mentioned are lined to the study.
3. Should be written a better description for the study area. It is useful for the reader of the manuscript to understand in which climate zone it is referring to. The article deals with climate change, rain trends and steamflow, therefore climate description is necessary.
4. Should be rewritten the paragraph “Data and methods”. What kind of rainfall data were used, eg monthly time series, by which stations. Also, to clarify if a homogeneity test has been done.
5. Line 137 …,“We compared the GPCP data with observed rainfall for the period 1981-2005” How this comparison was made? The GPCP data are grid data.
6. Line 146 “Some basic statically analysis (standard deviation, mean and coefficient of variation) was performed to assess the link between streamflow and rainfall variability and land cover/use dynamic” must be analysed.
7. Line 172 “In order to calibrate and validate the land cover classification the accuracy assessment was performed, and the kappa coefficient was used as the statistical parameter” Please provide references for this.
8. Line 212 “As changes in rainfall in the whole basin are not significant, our hypothesis is that the variations of streamflow described above are basically due to land use and land cover change.” This statement needs more documentation. In rainfall- streamflow and other rainfall characteristics play a role.
9. Line 281 “Between 2007 and 2014 (fig. 3f and fig. 3c), the trend….” Comparison is made between the two years it isn’t trend. The sentence must be rewritten.
10. Line 296 “Kossou station from 1981 to 1984 are not statistically significant at 95” from which this is deduced?
11. The image in Fig. 3 is too small, please enlarge and better quality if possible.
Author Response
Dear Reviewers,
Thank you again for the time you spent to read our work. Please find attached the responses to your concerned.
Best regards.

Reviewer 3 Report
The manuscript presents the changes of climate, land-use on hydro-power generation in West Africa. Although the topic of the manuscript is timely, unfortunately it contains serious deficiencies
including mainly the methodology. In my opinion the approach is overly simplified, there are factors that were not even mentioned in the manuscript that able to influence streamflow such as changes in the frequency of heavy rainfalls (the same amount of precipitation fall in shorter time period) or the effect of irrigation on the belowground discharge of the reservoir. I would suggest to set up a GIS based water balance model (such as Wetspass or other) for different time periods to calculate and compare the effect of changes of climate/land use.The introduction should be rewritten and better structured to become more readable and fluent, e.g separate sentences without logical connections among them. There are quite large number of unnecessary sentences stating obvious facts (see futher comments), thus the lenght of the manuscript could be reduced. Discussion should concentrate on the discussion of the study results with related relevant results of other studies with emphasis of the uncertainities of the results.
Further comments:
56-61: too much details (% numbers), only refer to these sources
61-64: I would put these sentences to the beginning of this paragraph
108: this aim -> the aim
117: increase -> increased
Figure 1. I suggest title: Location of Bandama basin
123-125: the power numbers are not really interesting, since they do not reflect if they are large
or small
Study area: please include more information about climate, soils, the land cover ratios, vegetation etc.
131-132: time resolution of the data? daily/monthly/annual?
134-137: "The GPCP products.." and "GPCP was successfully..." -> not relevant
138-139: these are results
142-146: "The non-parametric...time series." -> Not needed
146: statically -> statistical
151: for the years
160: land cover types -> land cover
169: growth of trees -> trees
172: For all mentioned past years (1988,2002,2016)? Please clarify.
174: compared
175 and 176-177 and later: in Bastola and Fran [34] etc.
177-184: could be reduced by referencing
184-187: results
191: nor clear what analysis was done (trend analysis p values?)
204: repeated content (L194)
206: express -> expressed
Table 2: P values do not tell much information about the trends (direction, magnitude), I would
present the observed trends for precipiation and streamflow of months and year.
Table 3: Not really interesting, I would show the mean data (perhaps with SD marks) on one graph
(prec+streamflow).
4.2.1. I would use only "land cover" (land use as build and agriculture areas is part of land
cover)
237: evergeen forest
242-243: repeated content
Table 5: I would also present here the trends of precipatation and sreamflow, not only the CV
266: cove -> cover
Figure 3: Axis captions should be improved, I would use the same axes settings (min/max values)
for all graphs to make more visible the differences.
274/282: rainfall intensity? I presume that it should be amount.
293-294: fragment?
369: "this trend increasing trend of" -> please revise
Author Response
Dear Reviewer,
Thank you again for the time you spent to make our paper acceptable. Please, find attached, the response to your concerned.
Best regards.

Round 2
Reviewer 2 Report
The manuscript has improved and all my objections answered by authors. I think it can be accepted for publication in the Journal.Author Response
Thank you very much for your usefull contribution to this work.
Reviewer 3 Report
The manuscript has developed greatly, thank you. I have only one issue: please include shortly that new GIS-based research is planned in the future to quantify water balance components of the basin in the discussion or in the conclusion.
Author Response
We agree. Thank you very much for this suggestion.
We have added in our perspectives the need to use GIS-based model to quantify water balance component in the “Discussion” and “Conclusion” sections of this revised version.